# Functional Characterization and Synthetic Application of Is2-SDR, a Novel Thermostable and Promiscuous Ketoreductase from a Hot Spring Metagenome

**DOI:** 10.3390/ijms232012153

**Published:** 2022-10-12

**Authors:** Erica Elisa Ferrandi, Ivan Bassanini, Susanna Bertuletti, Sergio Riva, Chiara Tognoli, Marta Vanoni, Daniela Monti

**Affiliations:** 1Istituto di Scienze e Tecnologie Chimiche “Giulio Natta”, Consiglio Nazionale delle Ricerche, Via Mario Bianco 9, 20131 Milano, Italy; 2Pharmaceutical Sciences Department, University of Milan, Via Mangiagalli 25, 20133 Milano, Italy

**Keywords:** enzyme discovery, biocatalysis, metagenomics, steroids, ketoreductases, stereoselectivity, functional characterization

## Abstract

In a metagenome mining-based search of novel thermostable hydroxysteroid dehydrogenases (HSDHs), enzymes that are able to selectively oxidize/reduce steroidal compounds, a novel short-chain dehydrogenase/reductase (SDR), named Is2-SDR, was recently discovered. This enzyme, found in an Icelandic hot spring metagenome, shared a high sequence similarity with HSDHs, but, unexpectedly, showed no activity in the oxidation of the tested steroid substrates, e.g., cholic acid. Despite that, Is2-SDR proved to be a very active and versatile ketoreductase, being able to regio- and stereoselectively reduce a diversified panel of carbonylic substrates, including bulky ketones, α- and β-ketoesters, and α-diketones of pharmaceutical relevance. Further investigations showed that Is2-SDR was indeed active in the regio- and stereoselective reduction of oxidized steroid derivatives, and this outcome was rationalized by docking analysis in the active site model. Moreover, Is2-SDR showed remarkable thermostability, with an apparent melting temperature (T_M_) around 75 °C, as determined by circular dichroism analysis, and no significant decrease in catalytic activity, even after 5 h at 80 °C. A broad tolerance to both water-miscible and water-immiscible organic solvents was demonstrated as well, thus, confirming the potential of this new biocatalyst for its synthetic application.

## 1. Introduction

Alcohol dehydrogenases (ADHs) are ubiquitous enzymes with a wide variety of substrate specificities, which are involved in several physiological processes that are essential for life and cell function. Moreover, ADHs, frequently indicated as ketoreductases (KREDs), represent one of the most important enzyme classes for the sustainable industrial preparation of optically pure *sec*-alcohols, e.g., active pharmaceutical ingredient (APIs) intermediates, from the corresponding ketones in stereoselective biocatalyzed processes [1,2].

Among others, nicotinamide (NAD(P)(H)) cofactors are the most used coenzymes in ADH-catalyzed oxidation/reduction of alcohol/ketone substrates [1,3,4]. Depending on the possible presence of metal ions in their active site and on the protein subunits’ length, NAD(P)(H)-dependent ADHs are mainly classified into two major groups: the zinc-dependent medium-chain dehydrogenases/reductases (MDRs) superfamily and the short-chain dehydrogenases/reductases (SDRs) superfamily, whose members do not require the presence of metallic ions for activity [5]. Other enzymes capable of performing carbonyl reduction reactions belong to the aldo–keto reductases (AKRs) superfamily, and are distinguishable from MDRs and SDRs by a (α/β)_8_-barrel fold and the lack of a typical Rossmann fold NAD(P)(H)-binding motif [6].

Various ADHs from extremophilic, e.g., (hyper)thermophilic, microorganisms have been discovered and characterized to some extent [7]. These enzymes show great potential for possible industrial applications since they frequently possess a superior stability at high temperatures, as well as in the presence of organic solvents and other protein denaturants [8].

Among the different possible approaches leading to the discovery of novel useful biocatalysts, metagenomics, i.e., the application of culture-independent methods for the identification of target sequences coding for the enzymes of interest in environmental DNA, is currently considered one of the most promising, especially when dealing with samples from extreme environments [9,10,11]. During the last years, the exploitation of metagenomics has indeed permitted the finding of thermostable and synthetically valuable biocatalysts from different enzyme classes, e.g., hydrolases [12,13,14], oxidoreductases [15,16] and amine transferases [17,18].

Recently, while searching for novel hydroxysteroid dehydrogenases (HSDHs), NAD(P)(H)-dependent oxidoreductases applicable in the selective oxidation/reduction of the hydroxyl/keto groups of steroids and bile acids [19], a new SDR sequence, named Is2-SDR (GenBank: QNN88924.1) and showing some phylogenetic relationship with known 7α- and 12α-HSDHs (Figure 1a), was identified in a metagenome prepared from an Icelandic hot spring sample [20,21]. However, when analyzing the sequence alignment results, the percent identity values of Is2-SDR with different HSDHs were modest (<40%), among the highest those obtained with a 7α-HSDH from *Bosea* sp. and with a 12α-HSDH from *Lysinibacillus sphaericus* (38% and 36% identity, respectively, Figure 1b). Nonetheless, other HSDH sequences, e.g., 3α-HSDHs and 3β-HSDHs, aligned with relatively close identity values, thus, making a direct inference of Is2-SDR substrate specificity from sequence analysis quite difficult (see Appendix A for the full set of data).

Subsequently, the preliminary characterization of Is2-SDR after recombinant production in *Escherichia coli* demonstrated that, although showing NADPH-dependent reductase activity toward different ketones, this new enzyme could not be classified as a HSDH since it did not accept any of the tested bile acids as substrates [20].

Nonetheless, very interesting results were obtained when using Is2-SDR in the biocatalyzed reduction of selected bulky substrates. Specifically, this novel biocatalyst permitted a kinetic resolution of the so-called Wieland–Miescher ketone, a useful intermediate in the synthesis of various complex natural products [22], as well as the stereoselective reduction of the bulky–bulky ketone 6-gingerol [23] (see the results and discussion sections for further details).

In this work, we report about the functional characterization of Is2-SDR, which permitted us to further highlight the broad substrate scope and outstanding stability of this new biocatalyst. Moreover, a promiscuous reductive activity toward oxidized steroid derivatives was demonstrated and rationalized by docking analysis on the protein 3D structural model.

## 2. Results

### 2.1. Functional Characterization of Is2-SDR

The recombinant production of Is2-SDR in *Escherichia coli* was carried as previously described [20]. Briefly, *E. coli* BL21(DE3) cells containing the expression plasmid with the synthetic gene coding for Is2-SDR were grown at 37 °C and 220 rpm until OD_600_ was 0.4–0.6; then, enzyme production was induced by the addition of *iso*-propyl β-D-thiogalactopyranoside (IPTG, 1 mM final concentration) and the bacterial culture was incubated at 200 rpm and 30 °C for 24 h. After the cells’ recovery and lysis, the cell extract was submitted to IMAC affinity chromatography and the C-terminal His-tagged Is2-SDR was recovered in homogeneous and soluble form with about 100 mg L^−1^ yields (see the materials and methods for details).

#### 2.1.1. Spectrophotometric Screening of Is2-SDR Reductive/Oxidative Activity in the Presence of Different Substrates

Preliminary spectrophotometric assays were set up with purified Is2-SDR at 340 nm to assess the NADP(H)-mediated oxidoreduction of different hydroxyl or carbonyl compounds (Figure 2). The analyses showed that this enzyme acts mostly as a ketoreductase with scarce capability in alcohol oxidation. Among the tested substrates, the highest specific activity (U mg^−1^) value was observed in the reduction of ethyl 3-methyl-2-oxobutyrate (**5**, EMOB, 8.18 U mg^−1^), while in the best cases, e.g., sulcatol (**8**) and cyclohexanol (**10**), much lower values (around 0.05 U mg^−1^) could be reached in the oxidation reactions.

#### 2.1.2. Influence of pH, Temperature, and Organic Solvents on Is2-SDR Activity and Stability

To evaluate the performances of Is2-SDR under modified reaction conditions, functional enzyme characterization was carried out by investigating the influence of different pH values and temperatures, as well as the presence of different organic solvents, on the biocatalyst activity and stability. The tests were performed at least in duplicate by spectrophotometric assays using 2 mM EMOB as a substrate and 0.2 mM NADPH as a cofactor (Appendix A).

Is2-SDR showed a quite broad pH activity profile in the tested range (pH 6.0–9.5), with a maximum at pH 6.5 (Appendix A).

Enzyme stability at different pH conditions was evaluated in the same range after the dilution of the Is2-SDR purified solution in 50 mM potassium phosphate buffer (PB) at different pH values. Activity assays were performed immediately after the dilution and at scheduled times (1, 3, 5 and 24 h). As shown in Figure 3a, at its activity maximum (pH 6.5), the biocatalyst was stable up to 5 h and the residual activity was about 60% after 24 h. A slightly superior stability was observed at pH 6.0, while at alkaline pHs, both enzyme activity and stability were progressively reduced and at pH 9.0 Is2-SDR was almost inactivated during the first incubation hour.

The temperature optimum was determined by standard spectrophotometric assays performed with the incubation of the cuvettes in a thermostatic bath at the needed temperature before the analysis. Is2-SDR showed a thermophilic character, with a maximum at 70 °C (Appendix A). Consistently, Is2-SDR also showed a remarkable thermostability, being quite stable (up to 5 h) when heated at 80 °C (Figure 3b).

The thermal stability of Is2-SDR was investigated also by circular dichroism (CD) analysis performed in far UV (250–190 nm) to analyze possible variations of the enzyme secondary structure depending on different temperatures (25, 80, and 90 °C). As shown in Figure 4, the obtained spectra were very similar, thus, confirming the thermostability of Is2-SDR. This result was also validated by the estimation of the apparent melting temperature (T_M_) of Is2-SDR. CD scans were performed every 5 °C and the variations of signal in Mdeg were registered at 220 nm, corresponding to the signal of β-sheets. The apparent T_M_ resulted to be around 75 °C, a value consistent with the previously presented stability data.

In order to evaluate the influence of water-miscible organic solvents on Is2-SDR activity and stability, the purified enzyme solution was properly diluted with PB, pH 7.0, then selected solvents, i.e., methanol (MeOH), ethanol (EtOH), dimethyl sulfoxide (DMSO), and acetonitrile (ACN), were added at concentrations of 5, 10, and 20% (*v*/*v*). After 1, 3, 5, and 24 h of incubation, the residual enzymatic activity was determined by spectrophotometric assays (Figure 5a). An overall good tolerance was observed and, in some cases, an increase in activity in the first hours of exposure, particularly in the case of 5% (*v*/*v*) MeOH, 10% (*v*/*v*) EtOH, and 10% (*v*/*v*) ACN, was observed. The best results in terms of stability at 24 h (about 50% residual activity) were achieved in the presence of 20% (*v*/*v*) MeOH, 20% (*v*/*v*) EtOH, and 10 or 20% (*v*/*v*) ACN.

Further investigations were carried out in biphasic systems set up by adding to the enzyme solution an equivalent volume of a water-immiscible solvent, specifically toluene, ethyl acetate (EtOAc), petroleum ether (Etp), *tert*-butyl methylether (tBME), and cyclopentyl methylether (CPME). After 1, 3, 5, and 24 h of incubation in a thermoshaker at 25 °C and 100 rpm, the samples of the respective aqueous phase were collected, and the residual enzymatic activity was assessed by standard spectrophotometric assays. As shown in Figure 5b, Is2-SDR showed an overall very good stability by retaining about 60% of the starting activity after 24 h of exposure to the tested solvents.

### 2.2. Screening of Bulky Ketones, α- and β-Ketoesters, and α-Diketones: Activity and Selectivity

As mentioned in the introduction, Is2-SDR previously demonstrated interesting reduction activity toward some sterically hindered ketone substrates, e.g., the Wieland–Miescher ketone (Table 1, **26**) [22] and 6-gingerol (**28**) [23].

The screening of Is2-SDR performances in reduction reactions was, thus, extended to a diverse series of carbonyl compounds. Table 1 includes both new and previously reported results to give a clear and complete picture of the biocatalyst substrate scope and, when applicable, stereoselectivity.

The reactions were run in an analytical scale (a 10 or 20 mM substrate in a 1 or 2 mL final volume, 10% (*v*/*v*) DMSO as cosolvent) exploiting the BmGDH (glucose dehydrogenase from *Bacillus megaterium*)/glucose system for NADPH cofactor regeneration (see the materials and methods for all the details). Conversions and enantiomeric/diastereomeric excesses (e.e./d.e.) were determined—according to the specific literature references—by GC, HPLC, and/or NMR analyses after extraction with EtOAc.

Benzylic carbonyl groups were considered first. Benzaldehyde (**17**) gave the best result in terms of conversion (>99%), while acetophenone (**18**) was converted into *S*-1-phenylethanol with very good e.e. (91%) and conversion. Interestingly, under the same reaction conditions, the symmetric bulky–bulky ketone benzophenone (**19**) gave only 25% conversion into the corresponding secondary alcohol.

The reduction of the 1,2-ketoesters methyl 2-oxo-2-phenylacetate (**20**) and methyl 3-methyl-2-oxobutanoate (**21**) resulted in quantitative conversions and good (82%, *R*-enantiomer) to excellent (99%, *R*-enantiomer) e.e.s, respectively. Instead, when testing the 1,2-diketone 1-phenylpropane-1,2-dione (**22**) as a substrate, a complex mixture of regioisomeric alcohols (the reduction of the keto groups in position 1 or 2) was obtained, which was characterized in terms of conversions by means of ^1^H-NMR. Specifically, 1-hydroxy-1-phenylpropan-2-one (PAC) was obtained as major product (67%), its 1-keto-2-OH regioisomer represented the 25% of the mixture, while the diols were less than the 10% of it. Unfortunately, the two regioisomeric alcohols appeared as inseparable by flash column chromatography (one spot in all the conditions tested) and, to the best of our efforts, also by chiral GC and HPLC (the direct and reverse phase were tried). Thus, it was impossible to determine their enantiomeric excesses.

Coming to polycyclic compounds, the bicyclic aromatic hydrocarbon α-tetralone (**23**) was converted into the corresponding *R*-tetralol with excellent conversion and e.e. (93% and 98%), whereas the β-tetralone (**24**) reduction resulted in a very low conversion and e.e. (<25%, *R*-enantiomer).

As previously reported, racemic *trans*-decalone (**25**), an aliphatic bicyclic system, was efficiently converted into the corresponding alcohol [20]. Two diastereomers were obtained: the *cis*-OH (with respect to the Cα-H) with >99% e.e. and the *trans*-OH isomers (e.e. = 62%). Like in the case of **25**, Is2-SDR was tested for its ability to discriminate between the two enantiomers of a racemic Wieland–Miescher ketone (**26**), a structurally more complex aliphatic bicyclic ketone, potentially leading to its stereospecific reduction or kinetic resolution. Is2-SDR proved to be able to catalyze the kinetic resolution of the racemic **26**, leading to the formation of the 4a*S*,5*S*-OH isomer (e.e. = 91%) and to the recovery of the 4a*R* enantiomer of **26** with 96.5% e.e. (see ref. [22] for details).

Finally, zingerone (vanilla acetone, **27**), a bioactive component of ginger extracts, was reduced by Is2-SDR with a poor conversion (22%) and a modest e.e. of 83% in favor of the *S*-enantiomer. At variance, 6-gingerol ((5*S*)-5-hydroxy-1-(4-hydroxy-3-methoxyphenyl)decan-3-one, **28**), was transformed into the 3*R*,5*S* diol (gingerdiol) with an excellent e.e. (>99%) and modest conversions (50 %) [23]. Interestingly, when testing ethyl 2-oxo-4-phenylbutanoate (**29**), a bulky–bulky 1,2-ketoester, as a substrate, the *R*-enantiomer of the corresponding 2-hydroxy ester was obtained with outstanding e.e. and conversion (>99 %).

### 2.3. Promiscuous Catalytic Activity of Is2-SDR in the Regio- and Stereoselective Reduction of Oxidized Bile Acids/Steroids

Given the reduction capability and the broad substrate scope of Is2-SDR, the possibility that specific steroidal compounds could be accepted as substrates by this biocatalyst was reconsidered. Indeed, as mentioned in the introduction, a classification of this enzyme as a typical hydroxysteroid dehydrogenase (HSDH) was not justified due to the lack of activity toward any of the tested bile acids (see Appendix A, for details). However, only oxidation reactions, e.g., using cholic acid as substrate, were considered in a previous work [20].

Selected oxidized bile acids were, thus, tested in spectrophotometric assays as the substrates of Is2-SDR-catalyzed reduction. No activity was detected in the presence of bile acids carrying keto groups only at positions 7 and/or 12, while substrates with a keto group at position 3, e.g., dehydrocholic acid (**30**, Figure 6), or the 3-keto and 3,7-diketo derivatives of cholic acid (Appendix A), could be reduced by Is2-SDR, although with quite low specific activities (25, 5 and 7.5 mU mg^−1^, respectively).

Dehydrocholic acid (**30**), with three ketones located at the 3, 7 and 12 positions, was further considered to better define Is2-SDR regio- and stereoselectivity. A small-scale preparative reaction (see experimental details in materials and methods) was conducted using, again, the BmGDH/glucose system for NADP^+^ regeneration and working at pH 8.0 (100 mM PB buffer) to permit good substrate solubility. After assessing the complete conversion of **30** into a more polar product by TLC, a full ^1^H- and ^13^C-NMR characterization allowed us to define Is2-SDR as capable of reducing the 3-keto position of this bile acid forming a β-configured secondary alcohol. Accordingly, the CHOH signal at 3.79 ppm appeared as a broad triplet with J_vic_ of ca. 2.7 Hz characteristic in a six membered ring of an equatorial hydrogen, thus, indicative of a β-oriented hydroxyl group, coupled with two adjacent methylene groups. Moreover, the spectrum obtained with the proton–carbon heteronuclear experiment (HMBC) indicated that the signals at 212.5 and 210.4 belonged to the CO-12 and CO-7 carbonyls since they showed long range correlations with the protons CH_3_-18/CH_2_-11 and CH_2_-6/CH-8, respectively (for full NMR spectra, see Appendix A).

5α-Dihydrotestosterone (**31**, Figure 6) was then used as a substrate, aiming to reduce its 3-keto position working this time in a biphasic medium, considering the poor water solubility of steroids. Accordingly, the enzymatic reduction of **31** was initially carried out in a 1:1 (*v/v*) mixture of *tert*-butyl methyl ether and PB buffer (pH 7.0, 50 mM) in the presence of Is2-SDR (1.0 U mL^−1^). Since BmGDH demonstrated a limited stability under these conditions, a formate dehydrogenase (FDH, 1.0 U mL^−1^)/ammonium formate (0.1 M) system [24] was employed for NADP^+^ (0.4 mM) regeneration and the reaction mixture was incubated at 25 °C and 180 rpm. To our surprise, no conversion was detected in TLC analysis even after 96 h. The same results were obtained using different biphasic systems (toluene, EtOAc, and petroleum ether as organic phase) regardless of the previously assessed Is2-SDR stability (see Figure 5b, EMOB-based assay of Is2-SDR stability in biphasic systems). Since **31** has not been investigated as a Is2-SDR substrate so far, control reactions in the same biphasic systems were performed using one of the best substrates of Table 1, α-tetralone (**23**). Even in this case, no α-tetralol formation was attested by GC analysis after 48 h, and just the 5% of **23** was converted after 96 h.

Thus, the biocatalyzed reduction of **31** was conducted in a monophasic system containing 10% (*v/v*) DMSO, in which the substrate was partially soluble. After product isolation by flash column chromatography, the nature of the more polar spot visible in TLC was assessed to be 5α-androstane-3β,17β-diol by ^1^H- and ^13^C-NMR characterization (Appendix A).

Finally, since 3β- and 17β-OH dehydrogenase activities are frequently associated in microbial HSDHs [25], androsterone (**32**), characterized by a 17-keto moiety, was tested as a substrate using a 40% (*v/v*) solution of DMSO in PB buffer as a reaction medium given its extremely low water solubility. After assessing the formation of a more polar product, which was isolated by means of a preparative TLC, a full ^1^H- and ^13^C-NMR characterization resulted in the identification of 5α-androstane-3α,17β-diol as the bioreduction product (Appendix A), thus, confirming the ability of Is2-SDR to act on both the 3 and 17 keto positions.

### 2.4. Homology Modelling of Is2-SDR Protein Structure and Active Site Docking Analysis

A 3D structural model of Is2-SDR was generated by using the SWISS-MODEL automated server [26]. The structure of the homo tetrameric 3-oxoacyl-[acyl-carrier protein]reductase from *Listeria monocytogenes* (Lm-FabG, PDB ID: 4JRO), belonging to the SDR superfamily [27], was selected as template among the 6963 templates found by the SWISS-MODEL server in the BLAST and HHBlits as it displayed the best combination of QSQE (Quaternary Structure Quality Estimate) and GMQE (Global Mean Quality Estimation) scores. The quality of the model was assessed by using Procheck and Verify 3D, tools for structural validation and analysis.

The overall folding topology of Is2-SDR is very similar to that of the other members of the SDR superfamily, exhibiting the Rossmann fold domain which core forms the cofactor binding site, and the so-called substrate binding loop, a crucial element for substrate recognition and enzyme function [28].

The superimposition of the Is2-SDR model with the Lm-FabG structure containing the NADP^+^ cofactor (Figure 7), along with sequence comparison (Appendix A), allowed the identification of the Is2-SDR amino acids involved in cofactor binding. As expected, most of these residues are highly conserved, such as Ile21 and Thr193 in the pyrophosphate binding region and the Tyr-X-X-X-Lys (Tyr158 and Lys162 in Is2-SDR) and Ile-X-Thr (Ile191 and Thr193 in Is2-SDR) motifs in the nicotinamide binding region [29]. Arg41, Thr42 and Ser18 interact with the O2′ ribose phosphate group of NADPH, consistent with the cofactor selectivity of Is2-SDR. The well conserved Tyr158/Lys162/Ser145 triad is involved in the enzyme catalytic function.

Among the various Is2-SDR substrates investigated in this work, dehydrocholic acid (DHCA, **30**, Figure 6) was selected for docking studies to get insight in the stereochemistry of its 3-ketone reduction catalyzed by Is2-SDR. Moreover, these studies also aimed at comparing the substrate-binding mode of this bile acid in Is2-SDR with the well-characterized binding mode of the bile acid 7-oxo-glycochenodeoxycholic acid (7-oxo-GCDA) included in the crystal structure of the 7α-HSDH from *E. coli* (Ec7α-HSDH, PDB ID: 1FMC). It is worth noting that the regio- and stereoselectivity of the reactions catalyzed by Is2-SDR and Ec7α-HSDH is different, as Is2-SDR reduces 3-keto bile acids producing 3β-OH derivatives, while Ec-7α-HSDH reduces 7-keto bile acids generating 7α-OH products.

Dehydrocholic acid was, thus, docked in the Is2-SDR active site and, as shown in Figure 8a, the orientation of DHCA in Is2-SDR active site was flipped compared to 7-oxo-GDCA in Ec7α-HSDH (Figure 8b), allowing the 3 keto group of DHCA to interact with the H of NADPH, Ser145, and Tyr158. As expected, while in Ec7α-HSDH the hydride of cofactor nicotinamide ring was positioned so as to attack the C7 ketone from the β-face of the C-O double bond plain resulting in the formation of an α-configured secondary alcohol, in Is2-SDR the hydride was positioned in a way that allowed it to attack the C3 ketone from the opposite prostereogenic face (α face) generating, instead, a β-configured hydroxyl group.

## 3. Discussion

The discovery of new extremozymes by exploiting metagenomics approaches paves the way to the development of sustainable biocatalytic industrial processes, which are highly foreseen in a green chemistry and circular economy view.

However, the lack of defined information about the microbial source and physiological role of metagenome-derived enzymes frequently poses some difficulties in envisaging their natural substrate specificity, as well as the possible influence of different reaction parameters on their catalytic performances.

These limitations are further amplified in the case of oxidoreductases belonging to the SDR superfamily, since automatic functional annotation of these biocatalysts is very often challenging due to the generally low sequence identity and high divergency within this protein family [30]. Moreover, some useful enzymatic features, e.g., tolerance to high temperatures and organic solvents, are still hardly predictable starting from the corresponding protein sequences.

In the case of Is2-SDR, the functional role and microbial origin of this enzyme could not be defined by simple databases search, the closest homologue (GenBank: MBC7327662.1; 90% identity at the amino acid level) being an uncharacterized oxidoreductase found in the metagenome-assembled genome of a bacterium isolated in an oil production facility [31]. It is worth mentioning that most of the (hyper)thermophilic ADHs characterized so far belong to the zinc-containing MDRs or the aldo–keto reductase (AKRs) superfamily [32,33]. A highly thermostable ADH from the SDR superfamily, possibly involved in carbohydrate metabolism, was found some years ago in *Pyrococcus furiosus* [34]; however, the similarity to Is2-SDR was not significative (about 33% sequence identity).

The first part of this work was, therefore, devoted to completing the functional characterization of Is2-SDR by extending the information about its substrate scope and selectivity, as well as about its performances under different reaction conditions.

A preliminary spectrophotometric screening of various alcohol and ketone substrates showed us a clear preference of Is2-SDR for reduction reactions instead of oxidations, along with a quite broad substrate scope, this fact confirming its applicative potential.

Is2-SDR also showed good activity in the quite broad pH range between 6.0 to 9.5, whereas its stability was generally higher at slightly acidic and neutral pHs when compared to alkaline pHs.

Coming to the effect of temperature on enzyme performances, Is2-SDR showed both a high thermophilicity (T_opt_ = 70 °C) and thermostability (T_M_ = 75 °C), these data being consistent with the environmental conditions of collection of the starting metagenomic DNA (hot spring, 85–90 °C, pH 5.0) [21]. An overall good stability was also observed when exposing Is2-SDR to both water-miscible and water-immiscible organic solvents.

The extended investigation about the synthetic performances of Is2-SDR in the biocatalyzed reduction of a set of different carbonyl compounds (Table 1) depicted a profile of a highly promiscuous ketoreductase showing interesting features in terms of substrate scope. Entries **17**–**19** highlight how the typical pattern of carbonyl groups possessing a sterically large and a small Cα groups is preferred by Is2-SDR, as both benzaldehyde (**17**) and acetophenone (**18**, [20]) were efficiently converted (with high e.e. in the case of the prochiral substrate **18**), while the symmetrically substituted, bulky–bulky ketone **19** resulted in a low conversion (25%) under the same reaction conditions.

Is2-SDR was found able to convert 1,2-ketoesters (entries **20** and **21**) with excellent conversions and e.e.s [20], while the enzymatic reduction of the 1,2-diketone 1-phenylpropane-1,2-dione (**22**) resulted in a complex mixture of regioisomeric alcohols, as this enzyme showed no specific regioselectivity for the 1 or 2 position of the substrate. No indications on the stereoselectivity in the reduction of **22** with Is2-SDR could be obtained as the outcome of the reaction could be analyzed only by the ^1^H NMR of the products mixture (see the material and methods section and the representative NMR spectra in the Appendix A).

Carbonyl group position resulted as crucial for Is2-SDR in the reduction of polycyclic compounds: α-tetralone (**23**) was in fact efficiently reduced (conversion and e.e. of 93% and 98%), whereas β-tetralone (**24**) reduction resulted in a very low conversion and e.e.. Further exploring polycyclic compounds with a higher degree of conformation freedom when compared to aryl–alkyl condensed systems, racemic *trans*-decalone (**25**) was found to be a good substrate for Is2-SDR, obtaining two different diastereomers with modest to very good e.e.s. [20]. Is2-SDR was also able to reduce the Wieland–Miescher ketone, a structurally complex aliphatic bicyclic ketone (**26**), catalyzing its kinetic resolution (see ref. [22] for details).

The substrate scope of Is2-SDR was further investigated with a focus on bulky or bulky–bulky ketones [35]. Zingerone (**27**), whose structural features could be at first regarded as a “small–large” case ketone, represents a peculiar substrate for Is2-SDR given the fact the “large” group—the substituted phenyl ring—is connected to the carbonyl group by a flexible ethylene chain. Is2-SDR reduced **27** with a low conversion and a modest e.e. [23]. Nevertheless, this enzyme was capable of converting the natural compound 6*S*-gingerol (**28**), a bulky–bulky substrate, which presents the same “large” group of **27** but is characterized by the presence of 1,3-hydroxyketone installed on the 6 and 8 positions (by traditional nomenclature) of a highly flexible decane chain, resulting in a modest conversion (50%) but in an excellent e.e. in favor of the 8*R*,6*S* diol [23]. Interestingly, substrate **29**, a 1,2-ketoester (formal) derivative of **27**, was efficiently reduced by Is2-SDR (conversion and e.e. >99%) highlighting, as demonstrated also by entries **20** and **21**, a preference towards this class of compounds.

In light of the broad substrate scope and the prevalent reductase activity shown by Is2-SDR, oxidized steroidal compounds were tested. Interestingly, a 3β/17β ketoreductase activity was demonstrated, thus, confirming the capability of this new biocatalyst of accepting sterically hindered substrates. This outcome could not be envisioned given the low sequence similarity with enzymes showing the same activity, i.e., 3β/17β-HSDHs (Appendix A).

Moreover, the specific activities of Is2-SDR toward the accepted steroids and bile acids were about three orders of magnitude lower than those usually observed with “true” HSDHs [20]; this finding further supporting the idea that these compounds are most likely reduced on the basis of Is2-SDR promiscuity. Additionally, at variance with the behavior of HSDHs in biphasic media [36], it is worth noting that Is2-SDR, although quite stable when incubated in the presence of different water-immiscible organic solvents, was incapable of reducing the tested steroids in such heterogeneous systems, thus, possibly suggesting a very low affinity toward these compounds [37].

In silico studies of the Is2-SDR structural model built using the Lm-FabG structure revealed a substantial similarity between Is2-SDR and the proteins of the SDR superfamily in terms of overall folding and conserved residues involved in cofactor binding and catalytic activity. The substrate-binding loop, that allows the correct substrate recognition, shows a helix–loop–helix structure, which is typical of the SDR superfamily.

The structure similarity between Lm-FabG and Is2-SDR suggests that these two enzymes could partially share substrate specificity. In fact, Lm-FabG is a β-ketoacyl-acyl carrier protein reductase that reduces 1,3-keto-thioesters and it is involved in lipid metabolism being required for elongation of C4 to C16 acyl chains [27]. Accordingly, Is2-SDR demonstrated to be active toward 1,2-keto esters, as stated above, and toward substrates containing long acyl chains, such as **28** and **29**.

Finally, the docking of the bile acid DHCA in the Is2-SDR active site clearly showed that this substrate, as experimentally observed, can be suitably positioned for the reduction of the 3-keto group to 3β-OH.

In conclusion, the extensive characterization of Is2-SDR carried out in this work highlighted the synthetic potential and the broad substrate scope of a novel, thermostable ketoreductase that also showed an interesting promiscuous activity towards steroidal compounds.

## 4. Materials and Methods

### 4.1. General

All reagents and solvents were purchased from either Merck (Darmstadt, Germany), TCI (Zwijndrecht, Belgium), or Fluorochem (Hadfield, UK) and used without further purification, unless otherwise stated. The glucose dehydrogenase from *Bacillus megaterium* (Bm-GDH) was recombinantly produced from *E. coli* BL21(DE3) cells harboring the plasmid pKTS-GDH, as previously described [38]. The NADP(H)-dependent FDH from *Pseudomonas* sp. 101 (Ps-FDH) was a kind gift from Prof. Tishkov (M.V. Lomonosov Moscow State University). The recombinant production of Ps-FDH was carried out by inoculating an overnight preculture (100 mL) in 1 L of LB_KAN30_ medium. The culture was maintained at 37 °C and 220 rpm up to an OD_600_ of 0.4–0.6; then, the expression was induced by the addition of 0.5 M IPTG (final concentration: 1 mM). After incubation at 17 °C for 72 h, the cells were harvested by centrifugation, and enzyme purification was carried out by immobilized metal affinity chromatography (IMAC), as described for Is2-SDR in the following.

### 4.2. Is2-SDR Expression and Purification

*E. coli* BL21(DE3) cells harboring plasmid pETite Is2-SDR were plated on a Petri dish containing LB_KAN30_ agar medium and incubated at 37 °C o/n. Then, a single *E. coli* colony was picked, pre-inoculated in 100 mL of LB_KAN30_ and incubated for 24 h at 37 °C and 220 rpm. After that, 20 mL of preculture was inoculated and grown in 1 L of LB_KAN30_ at 37 °C and 220 rpm up to an OD_600_ of 0.4–0.6. Once the desired OD_600_ value was reached, the expression was induced by the addition of 0.5 M IPTG to a final concentration of 1 mM, and the bacterial culture was incubated at 200 rpm and 30 °C for 24 h.

The cells were harvested by centrifugation (30 min, 4 °C, 5000 rpm) and resuspended in 20 mL of wash buffer (20 mM potassium phosphate buffer (PB), pH 7.0, 500 mM NaCl, 20 mM imidazole). Then, the lysis of the bacterial cells was conducted by ultrasonication (5 cycles, 30 s each at 40% of maximum power, Omni Ruptor 250-Watt Ultrasonic Cell Distrupter), followed by centrifugation (30 min, 4 °C, 10,000 rpm). Protein purification was performed using immobilized metal affinity chromatography (IMAC) with Ni-NTA Sepharose 6 Fast Flow (GE Healthcare, Milan, Italy) as stationary phase. Protein-soluble fraction was incubated on ice with Ni-NTA Sepharose resin (10 mL) for 90 min, and then loaded into a glass column (10 × 110 mm) connected to a Gilson Minipuls 3 peristaltic pump (flow rate: 1 mL min^−1^). After washing the resin with wash buffer, protein elution was carried out by increasing the concentration of imidazole, up to 300 mM.

Protein fractions were quantified by using the Bradford method, collected, and dialyzed o/n in 20 mM PB, pH 7.0, at 4 °C overnight. The presence of the target protein in the eluted fractions was verified by SDS-PAGE analysis.

### 4.3. Enzyme Assays and Is2-SDR Characterization

Is2-SDR activity was determined spectrophotometrically by monitoring the oxidation of NADPH or reduction of NADP^+^ at 340 nm (ε: 6.22 mM^−1^ cm^−1^) in the presence of a suitable substrate at room temperature using disposable plastic cuvettes. A total of 2–10 µL of 1:10 diluted enzyme solution was added to the assay solution containing 2 mM substrate and 0.2 mM cofactor in 50 mM PB, pH 7.0 (1 mL final volume), before measuring absorbance. The enzymatic unit is defined as the amount of enzyme that catalyses the conversion of 1 μmole of substrate in one minute under defined assay conditions.

Is2-SDR optimum pH, optimum temperature, stability at different pHs, temperatures, and cosolvents have been evaluated by the above-described assay using ethyl-3-methyl-2-oxobutyrate (EMOB, **5**, Figure 2) as a substrate. Assays were performed at least in duplicate, and results were compared to blanks.

To determine enzyme pH optimum, activity assays were performed in PB in a range of pHs between 6.0 to 9.0. Enzyme stability at different pH values was estimated by diluting 1:50 purified enzyme solutions in PB at different pHs (6.0–9.0) and incubating the diluted enzyme at room temperature. Activity assays were performed immediately after the dilution and at scheduled times (1, 3, 5, and 24 h).

To determine enzyme temperature optimum, activity assays were performed in a thermostatic spectrophotometer at temperatures between 20–90 °C. Once the instrument was set up at the desired temperature, a cuvette containing the substrate solution (2 mM in 50 mM PB, pH 7.0) was incubated in a thermostatic bath at the same temperature for 10 min. Then, 20 μL of 10 mM cofactor solution (NADPH in water) and 2–50 µL of 1:10 diluted enzyme solution were quickly added, and the cuvette was transferred to the instrument for the analysis. Enzyme stability at different temperatures was evaluated by diluting 1:50 the Is2-SDR purified solution in 50 mM PB, pH 7.0, in 1.5 mL tubes and incubating the solution at the desired temperature in a thermomixer. Activity assays were performed immediately after the dilution and at scheduled times (1, 3, 5, and 24 h).

The stability of Is2-SDR was evaluated in the presence of various organic solvents (methanol, ethanol, dimethyl sulfoxide, and acetonitrile) at different concentrations (5, 10, and 20 % (*v*/*v*)) and in biphasic systems in the presence of water-immiscible organic solvents (toluene, ethyl acetate, petroleum ether, *tert*-butylmethylether (tBME), and cyclopentyl methyl ether (CPME)). Specifically, Is2-SDR was diluted 1:50 with 50 mM PB, pH 7.0, and the organic solvent was added to reach the desired concentration, or an equivalent volume of a water-immiscible solvent was added to the diluted enzyme. The mixtures were incubated in a thermoshaker at 25 °C and 100 rpm, and activity assays were performed immediately and at scheduled times (1, 3, 5, and 24 h).

Bm-GDH and Ps-FDH activity assays were performed following the above-described assay procedure. For GDH, assay mixture was 50 mM glucose, 50 mM PB, pH 7.0, 0.2 mM NAD(P)^+^, while for FDH, assay mixture was 20 mM ammonium formate, 50 mM PB, pH 7.0, 0.2 mM NAD(P)^+^.

### 4.4. Bioinformatic and Protein Structure Analysis

Sequence alignments were carried out using standard protein Blast tool (https://blast.ncbi.nlm.nih.gov/, accessed on 1 April 2022). Phylogenetic trees were created using the Clustal Omega tool (http://www.ebi.ac.uk/Tools/msa/clustalo/, accessed on 1 April 2022) [39] and visualized with iTOL webserver (http://itol.embl.de/, accessed on 1 April 2022) [40]. Is2-SDR homology model was generated through the SWISS MODEL web server [26]. 4JRO corresponding to the 3-oxoacyl-[acyl-carrier protein]reductase (FabG) from *Listeria monocytogenes* was selected as the template to build up the homology model. The quality of the model was assessed by using PROCHECK [41] and Verify 3D tools [42] available on the UCLA-DOE LAB platform (https://saves.mbi.ucla.edu/, accessed on 1 May 2022). Protein visualization and ligand optimization were carried out using the Discovery Studio package 2021 (version 21.1.0 BIOVIA, San Diego, CA, USA). The conformers of the optimized ligand were generated using the Mercury module of the CCDC suite (https://www.ccdc.cam.ac.uk, accessed on 1 May 2022) and docked into the Is2-SDR active site using the GOLD module of the Hermes CCDC-suite [43].

### 4.5. Analytical Methods

CD spectra were recorded on a nitrogen-flushed Jasco J-1100 spectropolarimeter (Easton, MD, USA) interfaced with a thermostatically controlled cell holder. CD analyses were carried out in quartz cuvettes with 0.1 cm path length using purified Is2-SDR diluted in degassed water to a concentration of 0.15 mg mL^−1^. CD spectra were recorded in the range between 190 and 250 nm at 25, 80, or 90 °C, while for the determination of apparent melting temperature (T_M_), CD signal variations were detected at 220 nm using the following temperature program: 20 up to 65 °C at 5 °C min^−1^, data pitch each 2 °C, hold 30 s; 65 up to 90 °C at 2.5 °C min^−1^, data pitch each 0.5 °C, hold 30 s.

The NMR spectra were acquired in CDCl_3_, DMSO-d_6_, or in CD_3_OD, at room temperature (rt) on a Bruker AV 400 MHz spectrometer with a z gradient at 400 MHz for ^1^H-NMR analysis and 101 MHz for ^13^C-NMR.

Reactions were monitored via TLC (thin-layer chromatography) on pre-coated glass plates silica gel 60 with fluorescent indicator UV_254_ and treated with oxidizing solutions (phosphomolybdic reagent: (NH_4_)_6_MoO_4_ 42 g, Ce(SO_4_)_2_ 2 g, H_2_SO_4_ 62 mL, H_2_O 1 L; Komarowski reagent: 4-hydroxybenzaldehyde, 3 g; MeOH, 200 mL; H_2_SO_4_ 50%, 20 mL). At scheduled times, reaction samples were extracted with EtOAc and dried over Na_2_SO_4_, then submitted to chiral HPLC, GC, GC-MS, or NMR.

HPLC analyses were conducted using a Shimadzu LC-20AD high-performance liquid chromatography system equipped with a Shimadzu SPD-20 A UV detector. The samples, as EtOAc solutions, were analyzed on a Chiralcel OD (10 µm, 250 × 4.6 mm) through an isocratic elution (petroleum ether/isopropanol = 98:2), with detection at 210 nm. Retention times of ethyl 2-oxo-4-phenylbutanoate and *R*- and *S*-ethyl 2-hydroxy-4-phenylbutanoate were: 8.71 min, 16.02 min, and 10.45 min, which is in agreement with the literature reference data [44].

GC analyses were performed on an AGILENT 6850 (Net-work GC System) gas chromatograph equipped with a chiral capillary column (MEGA DEX DAC-BETA, Legnano, Italy; 25 m × 0.25 mm × 0.25 μm) and a flame ionization detector. At scheduled times, reactions samples (100 µL) were extracted with an equal volume of EtOAc and injected into the GC system. Different temperature gradients were applied to analyze the different compounds (according to the details reported below) using a general instrument setup: T_injector_ = 200 °C, T_detector_ = 250 °C, flow = 1 mL min^−1^, p= 100 kPa.

GC-MS analyses were performed using an Agilent HP-5MS column (30 m × 0.25 mm × 0.25 μm) on a Finnigan TRACE DSQ GC/MS instrument (Thermo Quest, San Jose, CA, USA). Inlet temperature: 280 °C; ion source temperature: 280 °C; MS transferline temperature: 280 °C.

### 4.6. Preparation of Standard Racemates

The chemical reductions of the substrates were performed following a standard protocol using NaBH_4_ as reducing agent. To a stirred solution of 0.15 M substrate (1 eq) in EtOH at 0 °C, NaBH_4_ (0.25 to 1 eq) was added. The reaction was maintained at 0 °C until the complete dissolution of NaBH_4_, then at rt for 4 h. After the indicated time, the reaction was quenched with a saturated solution of NH_4_Cl, then extracted with EtOAc (3×). Combined organic layers were dried over Na_2_SO_4_, concentrated in vacuo, affording the desired racemic alcohols that were used without any further purification. The racemic products were characterized by GC, HPLC, or NMR.

### 4.7. Analytical Scale, Biocatalyzed Reduction of Ketone Substrates with Is2-SDR

General screening protocol (final volume = 1–2 mL): BmGDH, NADP^+^, Is2-SDR and substrate (variable amounts depending on reduced compound, see details below), and 50 mM glucose were dissolved in a 10% (*v*/*v*) solution of DMSO in 50 mM PB, pH 7.0. Reaction mixtures were incubated at 25 °C, 180 rpm for 24–48 h. Conversions and enantiomeric excesses were determined by GC, HPLC, and/or NMR analyses after extraction with EtOAc.

#### 4.7.1. Enzymatic Reduction of **17**

Reaction mixture: 10 mM **17**, 2.0 U mL^−1^ BmGDH, 4.0 U mL^−1^ Is2-SDR, and 0.4 mM NADP^+^. TLC = DCM, phosphomolybdic reagent; GC = 100 °C for 30 min, T_injector_ = T_detector_ = 220 °C, flow = 1 mL min^−1^, p= 103 kPa *R_t_:*
**17**, 8.62 min; benzaldehyde, 3.47 min (Appendix A), in agreement with the literature reference data [45].

#### 4.7.2. Enzymatic Reduction of **18**

Reaction mixture: 10 mM **18**, 1.0 U mL^−1^ BmGDH, 4.0 U mL^−1^ Is2-SDR, and 0.4 mM NADP^+^. TLC = petroleum ether: EtOAc 9:1 (*v/v*), UV_254nm_. GC = 70 °C for 5 min, 2 °C/min until 120 °C, hold 8 min, 20 °C/min until 180 °C, hold 1 min, *R_t_*: **18** 13.98 min, *R*-1-phenylethan-1-ol 20.84 min, *S*-1-phenylethan-1-ol 21.34 min (Appendix A).

#### 4.7.3. Enzymatic Reduction of **19**

Reaction mixture: 10 mM **19**, 2.0 U mL^−1^ BmGDH, 4.0 U mL^−1^ Is2-SDR, and 0.4 mM NADP^+^. TLC = DCM: EtOAc 9:1 (*v/v*), permanganate solution. GC-MS = 60 °C, hold 1 min, 6 °C min^−1^ until 150 °C, hold 1 min, 12 °C min^−1^ until 280 °C, hold 5 min, *R_t_*: **19** 20.58 min, diphenylmethanol 20.64 min (Appendix A).

#### 4.7.4. Enzymatic Reduction of **22**

Reaction mixture: 10 mM **22**, 1.0 U mL^−1^ BmGDH, 1.0 U mL^−1^ Is2-SDR, and 0.4 mM NADP^+^. TLC = DCM:MeOH 98:2 (*v*/*v*), phosphomolybdic reagent. ^1^H-NMR = 1-phenylpropane-1,2-dione ppm, singlet at 2.00 ppm PhCOCOCH_3_; 1-hydroxy-1-phenylpropan-2-one, quartet centered at 5.25 ppm PhCH(OH)COCH_3_; 2-hydroxy-1-phenylpropan-1-one, singlet at 5.17 ppm PhCOCH(OH)CH_3_; 1-phenylpropane-1,2-diol, broad doublet centered at 4.36 ppm PhCH(OH)CH(OH)CH_3_ (Appendix A).

#### 4.7.5. Enzymatic Reduction of **23**

Reaction mixture: 10 mM **23**, 1.0 U mL^−1^ BmGDH, 4.0 U mL^−1^ Is2-SDR, and 0.2 mM NADP^+^. TLC = petroleum ether: EtOAc 9:1 (*v*/*v*), UV_254nm_. GC = 120 °C for 20 min, *R_t_*: **23** 9.28 min, *R*-α-tetralol 10.49, *S*-α-tetralol 12.17 min (Appendix A), in agreement with the literature reference data [46].

#### 4.7.6. Enzymatic Reduction of **24**

Reaction mixture: 10 mM **24**, 1.0 U mL^−1^ BmGDH, 4.0 U mL^−1^ Is2-SDR, and 0.4 mM NADP^+^. TLC = petroleum ether: EtOAc 9:1 (*v*/*v*), UV_254nm_. GC = after derivatization as acetates (Ac_2_O, Py in EtOAc) 110 °C for 40 min, *R_t_*: **24** 31.96 min, *R*-β-tetralol acetate 27.71 min, *S*-β-tetralol acetate diphenylmethanol 28.64 min (Appendix A), in agreement with the literature reference data [47].

#### 4.7.7. Enzymatic Reduction of **29**

Reaction mixture: 20 mM **29**, 1.0 U mL^−1^ BmGDH, 4.0 U mL^−1^ Is2-SDR, and 0.2 mM NADP^+^. TLC = DCM, phosphomolybdic reagent, UV_254nm_. HPLC = method described in material and methods, *R_t_*: **29** 8.71 min, *R*-ethyl 2-hydroxy-4-phenylbutanoate 16.02 min, *S*-ethyl 2-hydroxy-4-phenylbutanoate 10.45 min (Appendix A), in agreement with the literature reference data [44].

### 4.8. Is2-SDR Biocatalyzed Reduction of Bile Acids and Steroids

#### 4.8.1. Enzymatic Reduction of Dehydrocholic Acid (**30**)

Dehydrocholic acid (**30**, 25 mg, 0.01 mmol) was dissolved (5 mM) in 2.4 mL of 100 mM PB, pH 8.0. To this solution, glucose (25 mM), NADP^+^ (0.4 mM), BmGDH (2.0 U mL^−1^), and Is2-SDR (3.0 U mL^−1^) were added. The obtained mixture was incubated at 25 °C for 24 h and monitored by TLC analysis (DCM: MeOH 9:1 (*v*/*v*), Komarovsky reagent). After the complete conversion of the starting material, the reaction mixture was taken to pH 4, and it was freeze-dried. Reduction product was recovered by taking up the solid residue with EtOAc and by in vacuo concentration. The sample was dissolved in deuterated DMSO, and the NMR spectra were acquired on a Bruker AV 400 instrument at 302 K. Since the spectrum is quite complex, only a partial analysis was possible. The proton and carbon assignments were based on the proton homonuclear experiment COSY and on the proton–carbon heteronuclear experiments HSQC and HMBC. The multiplicity of the carbon nuclei was determined by the DEPT 135 experiment, which allows to distinguish among quaternary (s), methyne (d), methylene (t), and methyl (q) carbons. ^1^H NMR (DMSO-D_6_), δ: 3.79 (1H, t br, J = 2.7 Hz, H-3), 2.98 (1H, dd, J_6ax,5_ = 9.3, J_6ax.6eq_ = 12.6, H-6ax), 2.96 (1H, t, J = 13.0, J = 13.0, H-8), 2.75 (t, J = 12.7, J = 12.7 Hz, H-11ax), 2.18 (1H, m, H-5), 2.06 (1H, td, J_11ax,9_ = 12.6, J_8,9_ = 12.6 and J_11eq,9_ = 5.1 Hz, H-9), 1.88 (1H, J = 12.4 and J = 5.1 Hz, H-11eq), 1.58 (1H, m, H-14), 1.68 (1H, dd, J = 12.8 and 2.4 Hz, H-6eq), 1.26 (3H, s, CH_3_-19), 0.98 (3H, s, CH_3_-18), 0.75(3H, d, J = 6.3 Hz, CH_3_-21); ^13^C NMR (DMSO-D_6_), δ: 212.5 (1C, s. CO-12), 210.4 (1C, s, CO-7), 174.8 (1C, s, COO-24), 63.8 (1C, d, C-3), 56.3 (1C, s, C-13), 51.8 (1C, d, C-14), 48.0 (1C, d, C-8), 45.3 (1C, d, C-17), 44.8 (1C, t, C-6), 44.5 (1C, d, C-9), 39.9 (1C, d, C-5), 38.4 (1C, t, C-11), 35.9 (1C, s, C-10), 35.0 (1C, d, C-20), 34.5 (1C, t), 31.1 (1C, t), 30.4 (1C, t, C-22), 28.7 (1C, t), 27.2 (1C, t), 26.9 (1C, t), 24.6 (1C, t), 22.7 (1C, q, CH_3_-19), 18.6 (1C, q, CH_3_-21), 11.4 (1C, q, CH_3_-18).

#### 4.8.2. Enzymatic Reduction of 5α-Dihydrotestosterone (**31**)

5α-Dihydrotestosterone (**31**, 10 mg, 0.03 mmol) was dissolved (5 mM) in a 10% (*v*/*v*) solution of DMSO in 6.8 mL of 50 mM PB, pH 7.0. To this solution, glucose (25 mM), NADP^+^ (0.4 mM), BmGDH (2.0 U mL^−1^), and Is2-SDR (3.0 U mL^−1^) were added. The obtained suspension was incubated at 25 °C for 96 h and monitored by TLC analysis (DCM: MeOH 95:5 (*v*/*v*), Komarovsky reagent). After extraction with EtOAc and anhydrification over Na_2_SO_4_, the crude product was purified by flash column chromatography [DCM: MeOH 95:5 (*v*/*v*)] to yield a reduction product whose structure was attributed to 5α-androstane-3β,17β-diol (5.2 mg, 0.02 mmol, 60%).

^1^H-NMR (CDCl_3_, 400 MHz) δ 3.56 (t, *J* = 8.6 Hz, 1H), 3.53–3.46 (m, 1H), 2.03–1.90 (m, 1H), 1.87–1.65 (m, 4H), 1.64–0.98 (m, 17H), 0.85 (s, 3H), 0.73 (s, 3H). Signals in accordance with the literature data [48].

#### 4.8.3. Enzymatic Reduction of Androsterone (**32**)

Androsterone (**32**, 20 mg, 0.07 mmol) was dissolved (5 mM) in a 40% (*v*/*v*) solution of DMSO in 14 mL of 50 mM PB, pH 7.0. To this solution, glucose (25 mM), NADP^+^ (0.4 mM), BmGDH (2.0 U/mL), and Is2-SDR (3.0 U mL^−1^) were added. The obtained mixture was incubated at 25 °C for 96 h and monitored by TLC analysis (DCM: MeOH 95:5 (*v*/*v*), Komarovsky reagent). After extraction with EtOAc and anhydrification over Na_2_SO_4_, the crude product was purified by preparative TLC [DCM: MeOH 95:5 (*v*/*v*)] to yield a reduction product whose structure was attributed to 5α-androstane-3α,17β-diol (8.4 mg, 0.03 mmol, 42.8%).

^1^H-NMR (CDCl_3_, 400 MHz) δ 3.97 (q, *J* = 2.4 Hz, 1H), 3.56 (t, *J* = 8.6 Hz, 1H), 1.77–1.68 (m, 1H), 1.64–1.26 (m, 21H), 0.72 (s, 3H), 0.66 (s, 3H). Signals in accordance with literature data [48].

## Figures and Tables

**Figure 1 ijms-23-12153-f001:**
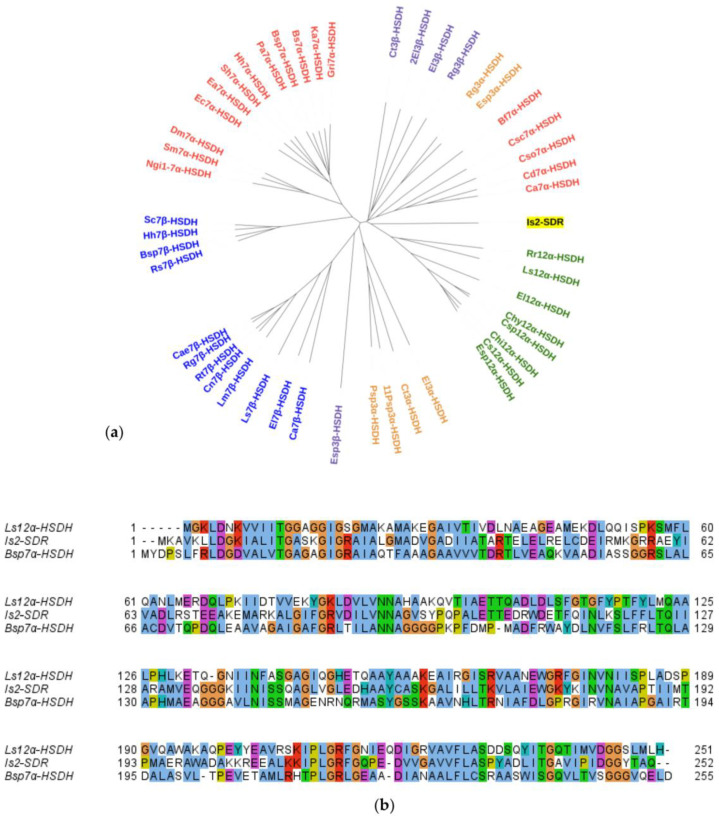
Sequence analysis of Is2-SDR. (**a**) Phylogenetic analysis of Is2-SDR with a set of HSDHs with different regio- and stereoselectivity (for details about the HSDH sequences see ref. [19]); (**b**) multiple sequence alignment of Is2-SDR with the 12α-HSDH from *Lysinibacillus sphaericus* (Ls12α-HSDH) and the 7α-HSDH from *Bosea* sp. (Bsp7α-HSDH).

**Figure 2 ijms-23-12153-f002:**
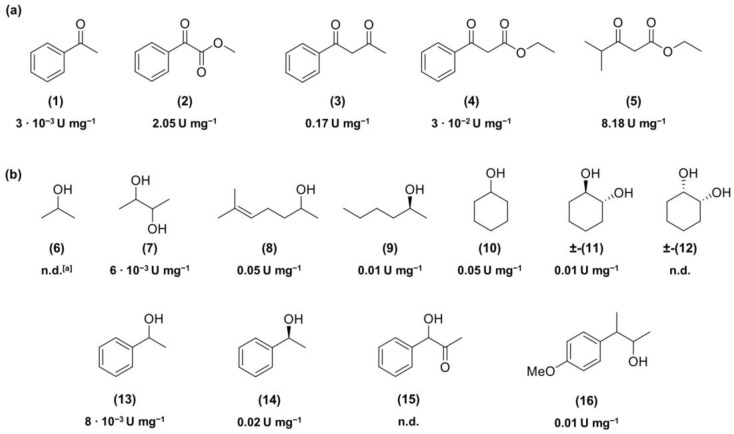
Substrates tested in spectrophotometric assays for the Is2-SDR-catalyzed reduction (**a**) or oxidation (**b**) in the presence of NADP(H) cofactor. Specific activity (U mg^−1^) values under standard conditions are reported. ^[a]^ n.d.: below detection limit.

**Figure 3 ijms-23-12153-f003:**
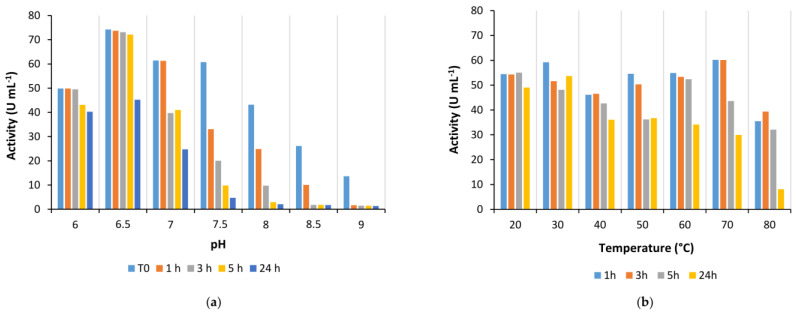
Stability of Is2-SDR at different pH values (**a**) and temperatures (**b**).

**Figure 4 ijms-23-12153-f004:**
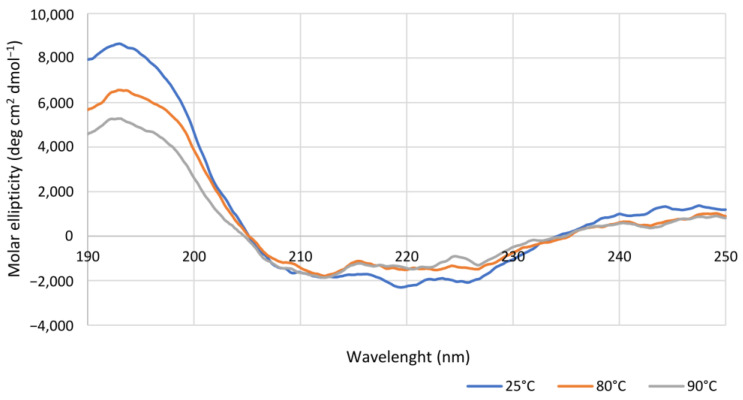
Overlapping of spectra obtained through Circular Dichroism spectroscopic analysis of purified Is2-SDR at 25 °C, 80 °C, and 90 °C (see materials and methods for experimental details).

**Figure 5 ijms-23-12153-f005:**
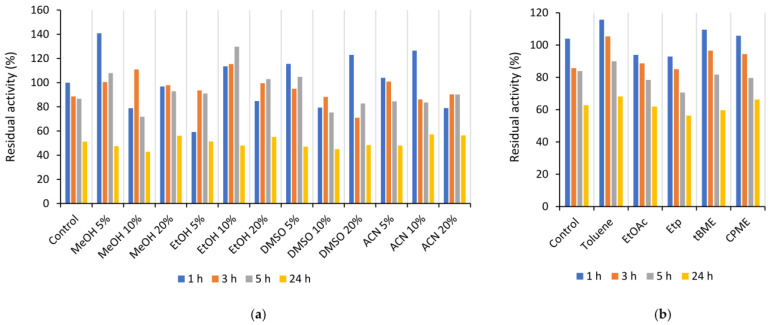
Stability of Is2-SDR in the presence of different organic solvents. (**a**) water-miscible organic solvents; (**b**) biphasic systems with water-immiscible organic solvents.

**Figure 6 ijms-23-12153-f006:**
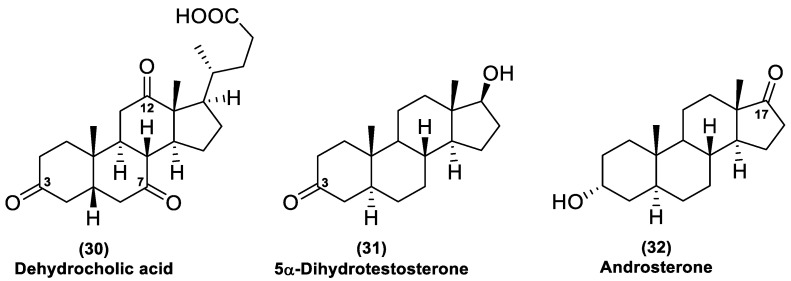
Bile acids and steroidal substrates tested in Is2-SDR-catalyzed reductions.

**Figure 7 ijms-23-12153-f007:**
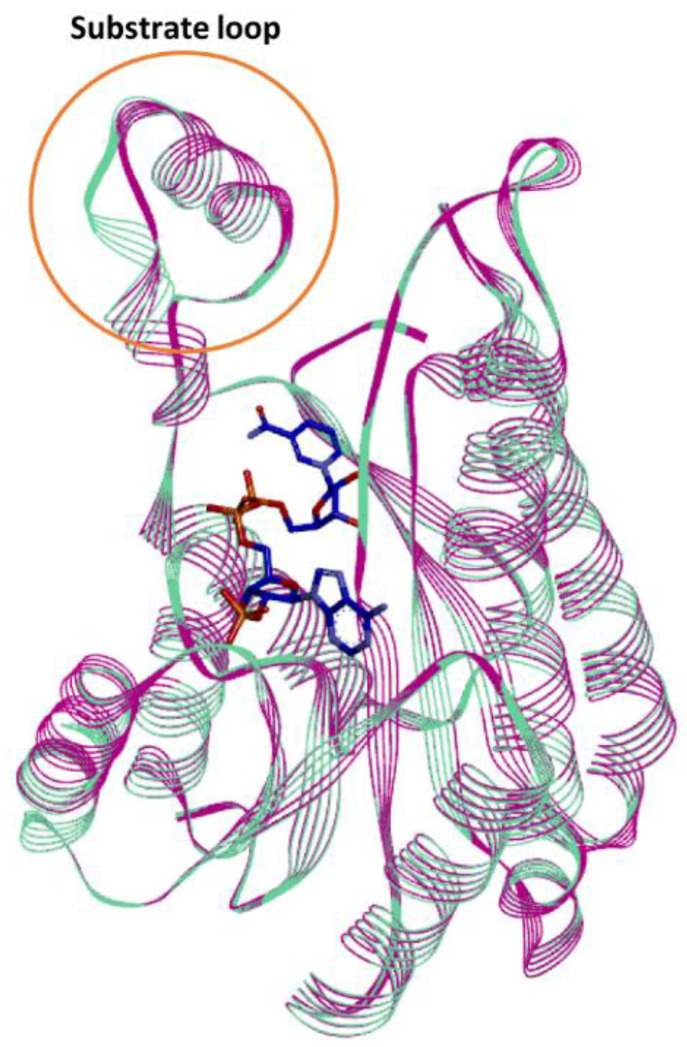
Superimposition of Is2-SDR (apo-form, green) with the holo-form of Lm-FabG (purple).

**Figure 8 ijms-23-12153-f008:**
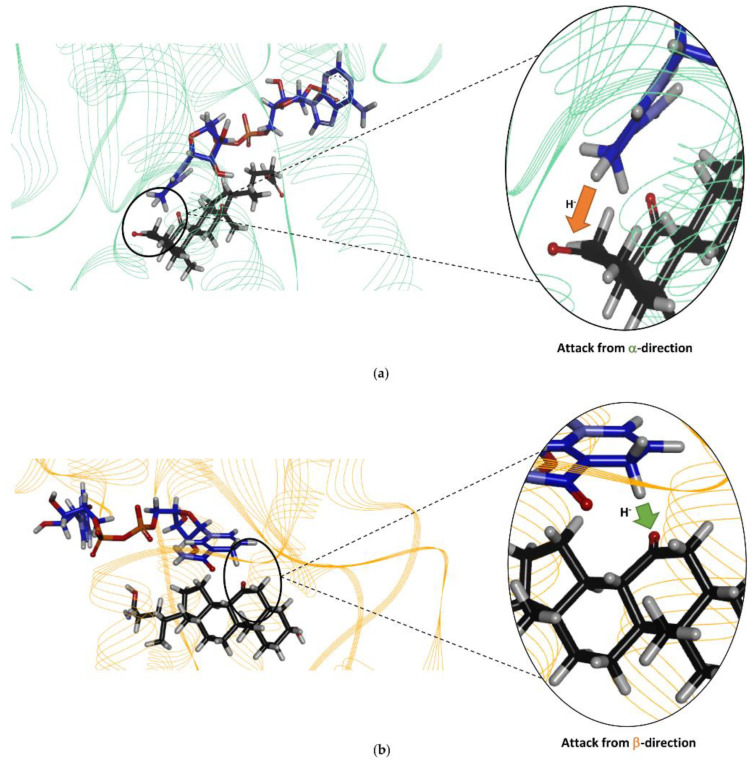
(**a**) Is2-SDR active site; (**b**) Ec7α-HSDH active site. The arrows indicate the direction of the cofactor hydride attack, NAD(P)H cofactor is shown in blue, the substrate (DHCA in (**a**) and 7-oxo-GDCA in (**b**), respectively) in black (see text for details).

**Table 1 ijms-23-12153-t001:** Substrate scope, conversion, and stereoselectivity in Is2-SDR-catalyzed reduction of different carbonyl compounds (chemical structure of products is shown in Appendix A).

	Substrate	Conversion (%)	Enantiomeric Excess (e.e., %)	Reference
**17**	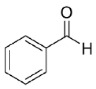	>99	--	This work
**18**	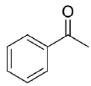	72	91.0 (*S*)	[20]
**19**	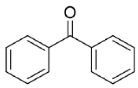	25	--	This work
**20**	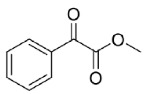	>99	82.0 (*R*)	[20]
**21**	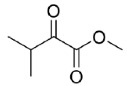	>99	98.4 (*R*)	[20]
**22**	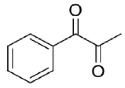	521-OH, 2-keto (PAC): 671-keto, 2-OH: 25Diols: 8	n.d. ^1^	This work
**23**	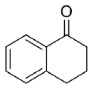	93	98 *(R)*	This work
**24**	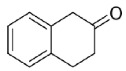	26	22 *(R)*	This work
**25**	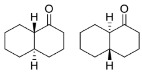	>99	*Cis*-OH: >99*Trans*-OH: 61.2	[20]
**26**	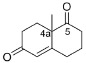	29 (4a*S*,5*S*-OH)32 (4a*R*,5S-keto) Traces of: (4a*R*,5*S*-OH) and (4a*R*,5*R*-OH)	91 (4a*S*,5S-OH)96.5 (4a*R*,5S-keto)	[22]
**27**	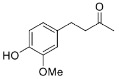	24	83 *(S)*	[23]
**28**	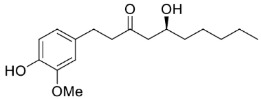	50	>99 *(R)*	[23]
**29**	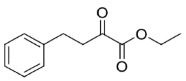	>99	>99 *(R)*	This work

^1^ n.d.: not defined. Racemic standards were prepared as reported in materials and methods.

## Data Availability

Not applicable.

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
