# Peer review of "Functional Characterization and Synthetic Application of Is2-SDR, a Novel Thermostable and Promiscuous Ketoreductase from a Hot Spring Metagenome"

_ijms, 2022, doi:10.3390/ijms232012153_

Round 1

Reviewer 1 Report

The paper under review deserves publication in the International Journal of Molecular Sciences. The topic is of interest for the journal readers and the described findings are scientifically sound. Overall, the paper is well written, there are few comments to further strengthen the paper:

1. Figure 1, the sequence of Is2-SDR started with KAVK, the starting residue Met was missing in the gene. Is this a full length or partial gene sequence?   

2. Figure 2, the unit activity of compounds 1 and 13,14 are similar, which suggesting the oxidoreduction are reversible. In fact, unit activity for compound 14 is 6 times better than 1, is this enzyme also favour oxidation of alcohol? Similarly, have the author tested the reverse reaction to convert ethyl 3-methyl-2-hydroxy butyrate to ethyl 3-methyl-2-oxobutyrate? 

3. A few studies had been reported ketoreductase enzyme from thermophilic bacteria with broad substrate specificity. For example, "Characterization of an aldo–keto reductase from Thermotoga maritima with high thermostability and a broad substrate spectrum". What make this enzyme standout from previous discovery?

Author Response

Replies to Reviewer's comments:

  1. Figure 1, the sequence of Is2-SDR started with KAVK, the starting residue Met was missing in the gene. Is this a full length or partial gene sequence?   

R: We thank the referee for noticing the lack of the starting Met in Figure 1, this amino acid was instead present as reported in the corresponding Genbank sequence and Figure 1 was corrected accordingly.

  1. Figure 2, the unit activity of compounds 1 and 13,14 are similar, which suggesting the oxidoreduction are reversible. In fact, unit activity for compound 14 is 6 times better than 1, is this enzyme also favour oxidation of alcohol? Similarly, have the author tested the reverse reaction to convert ethyl 3-methyl-2-hydroxy butyrate to ethyl 3-methyl-2-oxobutyrate?

R: We thank the referee for this comment and for her/his kind suggestion. It is true that, from our spectrophotometric assay, the specific activity of Is2-SDR towards compounds 1, 13 and 14 are very similar and this could suggest that the enzyme could promote oxidation comparably to reduction. However, the three activities are quite low when compared to those observed with other substrates, like 5 or 2. From our direct experience, Is2-SDR did not show any appreciable oxidative activity when exploited in analytic or preparative scale reactions followed by e.g. TLC, GC, HPLC and NMR, which always resulted in the recovery of the starting material. If oxidation products were formed, they were obtained in traces which could not be isolated.  

As far as the oxidation of 3-methyl-2-hydroxy butyrate concerns, this reaction was not conducted since it is known that SDRs and ADHs in general cannot oxidize 1,3-hydroxy ketones due to the stability of the six-membered ring that is formed intramolecularly via proton coordination operated by the carbonyl oxygen onto the hydroxyl hydrogen (acid-base complex according to Lewis). Details can be found in the literature (Bisogno F. R. et al. 2009, Org. Chem., 74, 1730-1732; Lavandera et al. 2008, Organic Letters,10, 2155-2158).

  1. A few studies had been reported ketoreductase enzyme from thermophilic bacteria with broad substrate specificity. For example, "Characterization of an aldo–keto reductase from Thermotoga maritima with high thermostability and a broad substrate spectrum". What make this enzyme standout from previous discovery?

R: It is indeed true that a few studies reports ketoreductase enzymes from thermophilic bacteria with broad substrate specificity , but Is2-SDR is of particular interest as it is active toward bulky-bulky substrates, i.e., sterically very demanding starting materials, such as compounds 19, 28, and 29. Bioreduction of bulky−bulky ketones has been defined as “an enormously challenging task due to the substrate steric hindrance and undesirable stereoselectivity” (Zhou et al. 2020 ACS Catal., 10, 10954−10966) and most of the ketoreductases active toward bulky-bulky substrate described to date are from mesophilic microorganisms, such as Kluyveromyces marxianus and Sporobolomyces salmonicolor. A comparison of Is2-SDR with the mentioned aldo–keto reductase from Thermotoga maritima is hard for different reasons, for example for the fact that the two enzymes belong to different families (i.e AKR and SDR) and for the lack of data about the activity of the T. maritima AKR toward bulky substrates.

Reviewer 2 Report

In the manuscript titled “Functional Characterization and Synthetic Application of Is2-SDR, a Novel Thermostable and Promiscuous Ketoreductase from a Hot Spring Metagenome”, Ferrandi et al reported a study on the functional characterization of the recently discovered short-chain dehydrogenase Is2-SDR. The enzyme is shown to be with broad substrate scope and high stability. The promiscuous reductive activity of the steroidal compounds was also explored. I recommend the publication of the study after minor revision.

1.     The authors should explain why the ee was only measured for a couple of selected substrates.

2.     In paragraph 2, on page9, the results of the control experiments are indeed surprising. The reason for the inactivity should be analyzed.

3.     Table1 on page8, the conversion of substrates 22 and 24 is low. Is that because of the smaller amount of enzyme used? For the other substrates, 4U/mL was used, while for substrates 22 and 24, 1U/mL was instead used.

4.     Line 606 on page 16, “Reaction mixture: 10 mM 19, 1.0 U mL-1 BmGDH......” should be “Reaction mixture: 10 mM 22, 1.0 U mL-1 BmGDH......”.

Author Response

Replies to Reviewer's comments:

  1. The authors should explain why the ee was only measured for a couple of selected substrates.

R: Enantiomeric excesses were measured for all the preparative scale reactions that promoted the formation of a chiral molecule, as reported in Table 1. For this reason, entry 17 and 18 do not report any indication of absolute stereochemistry. The only exceptions to this rule are the compounds of Figure 2, since reactions were conducted in the framework of a spectrophotometric assay to determine Is2-SDR specific activity on selected substrates, and for entry 22 of Table 1. The reduction of 1,2-phenylpropanedione, in fact, resulted in a complex mixture that was not separable by means of FC, GC and HPLC to the best of efforts.

  1.    In paragraph 2, on page9, the results of the control experiments are indeed surprising. The reason for the inactivity should be analyzed.

R: We thank the referee for suggesting this, we were also surprised by this result since we expected Is2-SDR to be active in biphasic systems given its high stability towards non-water miscible organic co-solvents (Figure 5b). We can thus hypothesize that the inactivity showed by the enzyme in the control experiments should be due to a problem of substrate partitioning between the water and the organic phase in conjunction with affinity issues: the low concentration of substrate that is reached in the water phase could not be enough to trigger the enzymatic transformation. Specific kinetics studies on this issue (evaluation of KM and kcat of the Michaelis-Menten kinetics) will be the focus of our up-coming investigations in which Is2-SDR will be investigated in non-traditional reaction media for tailored applications (like flow synthesis).

  1. Table1 on page8, the conversion of substrates 22 and 24 is low. Is that because of the smaller amount of enzyme used? For the other substrates, 4U/mL was used, while for substrates 22 and 24, 1U/mL was instead used.

R: We thank this referee for highlighting this point. Conversion of substrate 22 was kept low by choice since we did not want to obtain only diols, thus enzyme loading was lowered given Is2-SDR high activity on both the carbonyl groups of 1,2-phenylpropandione.

As far as compounds 24 concerns, the loading was a mistake of ours: as for the other compounds 4 U/mL were used. Text was amended accordingly.

  1. Line 606 on page 16, “Reaction mixture: 10 mM 19, 1.0 U mL-1 BmGDH......” should be “Reaction mixture: 10 mM 22, 1.0 U mL-1 BmGDH......”.

R: We thank the reviewer for noticing this mistake. The text was corrected accordingly.